# Underreporting of Obesity in Hospital Inpatients: A Comparison of Body Mass Index and Administrative Documentation in Australian Hospitals

**DOI:** 10.3390/healthcare8030334

**Published:** 2020-09-11

**Authors:** Alexandra L Di Bella, Tracy Comans, Elise M Gane, Adrienne M Young, Donna F Hickling, Alisha Lucas, Ingrid J Hickman, Merrilyn Banks

**Affiliations:** 1Department of Nutrition and Dietetics, Royal Brisbane and Women’s Hospital, Brisbane 4029, Australia; Adrienne.Young@health.qld.gov.au (A.M.Y.); Merrilyn.Banks@health.qld.gov.au (M.B.); 2Centre for Health Services Research, The University of Queensland, Brisbane 4067, Australia; t.comans@uq.edu.au; 3School of Health and Rehabilitation Sciences, The University of Queensland, Physiotherapy Department, Princess Alexandra Hospital, Centre for Functioning and Health Research, Metro South Health, Brisbane 4102, Australia; Elise.Gane@health.qld.gov.au; 4Department of Nutrition and Dietetics, The Prince Charles Hospital, Brisbane 4032, Australia; donna.hickling@health.qld.gov.au; 5Department of Health Information Services, Royal Brisbane and Women’s Hospital, Brisbane 4029, Australia; Alisha.Lucas@health.qld.gov.au; 6Department of Nutrition and Dietetics, Princess Alexandra Hospital, Brisbane 4102, Australia; Ingrid.Hickman@health.qld.gov.au

**Keywords:** obesity, body mass index, hospitals, hospital costs, health care costs, inpatients

## Abstract

Despite its high prevalence, there is no systematic approach to documenting and coding obesity in hospitals. This study aimed to determine the prevalence of obesity among inpatients, the proportion of obese patients recognised as obese by hospital administration, and the cost associated with their admission. A cross-sectional study was undertaken in three hospitals in Queensland, Australia. Inpatients present on three audit days were included in this study. Data collected were age, sex, height, and weight. Body mass index (BMI) was calculated in accordance with the World Health Organization’s definition. Administrative data were sourced from hospital records departments to determine the number of patients officially documented as obese. Total actual costing data were sourced from hospital finance departments. From a combined cohort of *n* = 1327 inpatients (57% male, mean (SD) age: 61 (19) years, BMI: 28 (9) kg/m^2^), the prevalence of obesity was 32% (*n* = 421). Only half of obese patients were recognised as obese by hospital administration. A large variation in the cost of admission across BMI categories prohibited any statistical determination of difference. Obesity is highly prevalent among hospital inpatients in Queensland, Australia. Current methods of identifying obesity for administrative/funding purposes are not accurate and would benefit from reforms to measure the true impact of healthcare costs from obesity.

## 1. Introduction

The prevalence of obesity has tripled over the past three decades in Australia, with the latest figures from 2017–2018 indicating that 31.3% of Australian adults are obese (body mass index [BMI] of≥30 kg/m^2^) [1,2,3] and a further 35.6% are overweight (BMI 25.00–29.99 kg/m^2^) [1]. Obesity is a leading contributor to the burden arising from chronic diseases such as cardiovascular disease, type 2 diabetes, and some cancers, which places significant financial strain on the healthcare system [2,4,5]. An estimation of the cost of Australian obesity has previously been based largely on population data, with the cost of obesity in Australia in 2011–2012 estimated to be AU$8.6 billion [6]. However, specific evidence is limited regarding the prevalence of obesity amongst inpatients of acute hospitals in Australia and the subsequent impact of obesity on the costs of those admissions. To our knowledge, the only study conducted within an Australian tertiary hospital of the point prevalence of obesity amongst inpatients reported a rate similar to local population norms of 25%, but with an over-representation of people classified as obesity class II and higher (BMI ≥ 35 kg/m^2^) [7]. Internationally, studies consistently report that a considerable proportion of healthcare dollars is spent on the treatment and management of obesity-related conditions; however, the direct cost of obesity on hospital admissions remains unclear [8,9,10].

Research conducted in the Australian healthcare system suggests there is an association between obesity and higher healthcare costs. An Australian population-based prospective cohort study of patients aged 45 to 79 years demonstrated that obesity and being overweight contributed to one in every eight hospital admissions and accounted for 17% of total dollars spent on hospitalization [10]. Increased BMI has been associated with increased cost of hospital admission, with costs for patients classified as obese class II and higher shown to be 50–100% higher than patients with a BMI within the normal range (BMI 18.5–25.0 kg/m^2^) [11]. Despite heterogeneity in the methodology of studies investigating the economic burden of obesity within healthcare globally, previously identified factors contributing to increased costs may include: purchase and availability of bariatric sized equipment; staff resources for mobilising and patient assistance; increased difficulty in some medical procedures; and increased length of stay [12,13,14,15].

Despite potential higher costs associated with obesity during acute admissions, there is no systematic approach to documenting and coding BMI-defined obesity in acute hospital settings. The Australian Refined Diagnosis Related Group (AR-DRG) is a classification system that uses specific AR-DRG codes to group inpatients based on similar clinical conditions that will require similar amounts of hospital resources for treatment and care provision. In assigning an AR-DRG, the patient’s case is assigned to one of 23 Major Diagnostic Categories (MDC) in order to cluster patients into medically meaningful groups. At this time, pre-MDC processing is performed to identify very high cost AR-DRGs. These identify the AR-DRGs with the highest clinical complexity. The case is then allocated to a partition within the MDC (medical, surgical, or other) before a Diagnosis Complexity Level (DCL) and Episode Clinical Complexity Score (ECCS) are calculated. Once this is complete, the AR-DRG is assigned to the patient.

DCLs refer to a complexity weight that is assigned to a diagnosis for a specific AR-DRG of a hospital admission episode while ECCS represents a cumulative score of the DCLs assigned to a patient episode ranging from 0 to 31.25 [16,17]. The higher the value of the ECCS, the more complex the case. It is important that this classification is performed accurately and systematically as hospitals rely on assigned AR-DRG codes to receive funding allocations from the Australian Government for the provision of care and hospital services to the respective admitted inpatients, referred to as Activity Based Funding [18]. Current practice for coding obesity, and consequently informing funding allocation, requires medical officers’ written documentation in patient records stating that obesity was a factor that impacted on patient care. This coding practice is irrespective of any BMI recorded by the treating team.

This project endeavoured to understand the significance of obesity within acute hospital admissions and how this is reflected in current coding practices in order to develop management strategies for activity based health care delivery and resource use. The primary aim of this study was to identify the point prevalence of obesity in a cohort of inpatients across three tertiary hospitals in Brisbane, Australia. Secondary aims of the study were to identify what proportion of patients with obesity were identified as obese by the hospital-based coding system and how the cost of hospital admissions compares across BMI categories.

## 2. Materials and Methods

This study was a multi-site cross-sectional audit of inpatients across three tertiary hospitals (Sites A, B, and C) in Brisbane, Australia in 2016. This study was approved by the Institutional Human Research Ethics Committee (HREC/16/QRBW/511) with a waiver of patient consent granted based on negligible risk. At Site A, height and weight data were collected over two consecutive days in December 2016. For Sites B and C, height and weight data from previous hospital-wide audits, which occurred on a single day between August and October 2016, were collated for the purpose of this analysis. Eligible patients were excluded if data was unavailable or incomplete.

Hospital inpatients admitted during the data collection period were eligible for inclusion in the study. All acute and sub-acute wards were included in data collection at Site B and C; however, sub-acute and mental health wards were excluded at Site A due to the off-site location of these wards. Mental health was also excluded at site B due to its off-site location. Across all sites, excluded patients were based on those deemed by the research team as not appropriate for data collection, including: maternity patients, palliative patients, medically unstable patients, and patients with low levels of consciousness deemed unable to participate. Patients with amputations were included, with weights adjusted before analysis using criteria to account for body portions as a percentage amputated [Adjusted weight = (current weight/100 − percentage of amputation) × 100] [19,20].

Age, sex, height, and weight were collected. Height documented in the patient’s medical record was used where available. Height and weight should be documented on hospital admission per local hospital procedures, with weight documented weekly thereafter, however in clinical practice, this is not always completed. If not documented in the patient’s medical chart, height was measured with a stadiometer. Where this was not possible, height was estimated by alternative measure (i.e., demispan/ulnaspan) or verbally reported by the patient. At all sites, weight documented in the patient’s medical record within one week of audit day was used where available. If not documented, weight was measured on the day of the audit using calibrated scales; where this was not possible, weight was reported by the patient or estimated by the investigator. Site A utilised final year undergraduate dietetic university students who were trained and supervised in pairs to collect data for all patients. At Sites B and C, data were obtained from hospital wide audits, which utilised dietitians and nursing staff trained as data collectors.

BMI was defined and categorised by WHO criteria (BMI > 30.0), with obese class I, II, and III all included in the overall definition of obesity [21]. The International Statistical Classification of Diseases and Related Health Problems 10th edition Australian modified (ICD-10AM) is used to categorise episodes of patient care into AR-DRGs; clinically coherent groups with similar resource utilisation for the purposes of case-mix based funding and analysis. If a medical officer documents “obesity” in a medical chart and there is evidence that the obesity met the Australian Coding Standard (ACS) for Additional Diagnoses (ACS 0002 Additional Diagnoses), an administrative officer will “code” that patient as being obese. To meet the ACS for “obese”, there needs to be evidence in the medical chart demonstrating that obesity affected patient management in terms of requiring any of the following: commencement, alteration or adjustment of therapeutic treatment; diagnosis procedures; or increased clinical care and/or monitoring. ACS does not refer to BMI to code obesity. In addition, coding a patient for obesity may not necessarily always change the allocated AR-DRG of the individual as this is dependent on the DCL assigned and other co-existing additional diagnoses. Additionally, if a patient is obese and is admitted with an unrelated health problem, supplementary diagnoses might be applied, which does not change the AR-DRG.

Costing data are routinely collected for all inpatients by the hospitals’ finance department. The total actual cost of the admission for patients included in the study was obtained from the finance departments. This is calculated at the individual patient level using AR-DRGs, length of stay, and cost of treatment (including emergency, intensive care and outpatient services, pathology, pharmaceuticals, imaging, procedures, and surgery). At Site A, data on additional diagnoses and the DCL weighting was obtained from hospital administration services in order to identify if the additional diagnoses for obesity changed the AR-DRG for that admission.

Statistical analyses were performed using Stata (Stata Statistical Software: Release 13. StataCorp LP, College Station, TX, USA). Summary statistics for continuous and categorical variables were produced for the whole cohort and by hospital site. Differences between sites for key clinical characteristics were evaluated with one-way ANOVA (continuous dependent variable) or chi-square tests (categorical dependent variable). Significance was set at *p* = 0.05. The evaluation of cost data was limited to the period of hospital admission associated with acute treatment, excluding possible subsequent sub-acute care such as rehabilitation. Cost data were found to be non-normally distributed on visual inspection of histograms, therefore data are presented descriptively with mean (SD), median, and interquartile range.

## 3. Results

There was complete data available for 1327 inpatients included in the audit across the three sites. At site A, 407 patients were eligible, however complete data were obtained for 341 patients (66 excluded). Site B had the largest cohort, with complete data for 565 patients out of a total of 704 patients (139 excluded). At site C, complete data were available for 422 patients, out of 494 eligible patients (71 excluded). The overall cohort was 57% male (*n* = 758) with a mean (SD) age of 60.7 (19.3) years and a mean (SD) BMI of 28.20 (8.50) kg/m^2^ (Table 1).

There was a significant difference between sites for age (*p* < 0.01) and sex (*p* = 0.04), but not BMI (*p* = 0.17). Figure 1 illustrates the prevalence of BMI categories within the combined inpatient cohort, with 30% (*n* = 397) classified as overweight and 32% (*n* = 421) classified as obese, including 7% of patients (*n* = 93) with a BMI > 40.00 kg/m^2^. There was no difference in prevalence of BMI categories between sites (Table 1). Eight percent (*n* = 83) were identified as underweight across all sites. These patients were not studied further in this research as it was outside the scope of the study.

ICD coding was available for 1278 patients (96% of the cohort). Of those with available coding data, half of the patients who were obese according to BMI were also coded by hospital information management as obese (*n* = 203 of 405, 50%). The proportion of the cohort that was coded for obesity and overweight according to BMI was 2% (*n* = 22). However, both of these results varied substantially across sites and BMI categories (Table 2). An obesity code was rarely incorrectly attributed to patients with a normal BMI (*n* = 1 Site B; *n* = 2 Site C).

Length of stay was variable both between sites and between BMI categories, with no clear trend to an increased length of stay attributable to obesity (see Table 3). Costing data were available for Sites A and C only. Total actual cost percentiles are presented in Table 3. While overweight and obese inpatients costed more on average than normal weight patients, this was not significant due to the large variation present, with some higher categories of obesity costing less on average than lower obesity categories.

At Site A, 103 patients were identified as obese (BMI ≥ 30 kg/m^2^). ICD coding data were available for 94 patients (91%). As this was the primary research site, we elected to run a further costing analysis on this cohort of patients. Of these patients, 25 (27%) had obesity recognised in coding. A secondary analysis was conducted to assess whether a code for obesity would have changed the DRG allocated for all patients with a BMI of ≥30 kg/m^2^ at Site A. Results demonstrated that if all 94 obese patients had an additional code of “obese”, only three (3%) would have had a change in DRG and subsequent activity based funding. It was noted at Site B that the length of stay was highest in the underweight patient group at a mean (SD) of 48.15 (67.00) days, however for the purpose of this obesity audit, it was not further investigated.

## 4. Discussion

This audit of three tertiary hospitals classified 32% of inpatients as obese. However, only half of patients classified as obese by BMI were coded as obese. This study highlighted the large variation in coding practices across the three sites. Given that coding of co-morbid obesity may affect the revenue indicated for an admission, this may indicate that hospitals may be missing out on revenue needed to manage patients with complex co-morbidities.

In order to understand the potential influence of regional factors on prevalence rates of obese patients within acute hospitals, the findings from the present study make for an interesting comparison to those of a study conducted in a tertiary hospital in Perth, Western Australia. Dennis and colleagues [7] performed a point prevalence study of BMI in 2015 at a single institution of a similar size to Site B in the present study. The prevalence of overweight and obese inpatients was 32.4% and 22.3%, respectively. These figures are similar to the prevalence of overweight patients in the present study (30%), however the prevalence of obese patients is higher, at 32%, in our Queensland sites. This is in keeping with population data that demonstrates a higher prevalence of obesity in the general Queensland population (30%) compared with the general Western Australian population (25%) [1]. In addition to a higher prevalence of adults with obesity, in comparison with Western Australia, Queensland has a higher proportion of adults with three or more long-term health conditions (44.7% vs. 40.4%), current daily smokers (16.1% vs. 14.3%), and adults who engage in no or low exercise levels (67.2% vs. 62.7%) [1].

A key finding of this study was the discrepancy between the number of patients classified as obese by BMI (32%) and the number of patients coded for obesity for administrative purposes (16%). At present, the participating hospitals do not use BMI in their decision to code a patient for obesity. Rather, this decision is based on the documentation of changes to a patient’s clinical management as a result of their obesity. Furthermore, any increase in allocated funding is case-specific, depending on the clinical condition, DCL, and weighting. It is assessed on an individual basis whether or not a simultaneous obesity coding changes the AR-DRG.

The coding practices at the participating hospitals are currently under review in 2020 and are likely to shift to a process that does incorporate BMI into the coding of obesity. Improved documentation of BMI as part of routine clinical practice and coding will have implications for the ability of large hospitals to monitor longitudinally the BMI of inpatients, on both the individual patient level and the broader hospital level. It is important that hospital staff responsible for undertaking height and weight measurements of patients on and during admission complete this task in order to avoid incomplete patient data, which could potentially negatively impact the funding allocation to treat and manage obesity. Activity based funding models function within complex health system regulations. Incorporation of coding systems to capture the impact of obesity on health care delivery could add value to current processes. While we found some indication that patients with obesity cost more on average compared with those who were of a normal weight, in contrast to previous research, we did not find evidence of greater health care costs in patients with greater levels of obesity. There may be several reasons for this finding. The number of patients that were categorised as obese class III was relatively small and therefore, with highly skewed cost data, it was not sufficient to make a reliable estimate. In addition, costing data is limited by the information available to the finance department. The additional costs of managing obese patients in terms of staff resources and equipment are not captured by current data systems, with average ward costs attributed to every patient. Further limitations of this study include the variation in data collection procedures between sites and the lack of costing data from one site due to significant transformation in records management at the time. Further limitations of this study include the variation in data collection procedures between sites and reliance on self-reported height and weight data in some circumstances and the lack of costing data from one site due to a significant transformation in records management at the time.

The results of the present study demonstrate that obesity is highly prevalent amongst acute hospital inpatients. Despite a steady trend over several decades of increasing rates of overweight and obese people in the Australian population [1,22], there is no consistent or systematic way to routinely collect data to measure the prevalence or impact of obesity on the healthcare system. Given the differences in the frequency of coding obesity between the three hospitals in this study, it appears that the current definition is not robust. At present, ICD coding of obesity cannot be used to track prevalence or understand the impact on the hospital system because it is not sufficiently accurate (i.e., a true reflection of BMI). In order to better manage systems for future demand, it is important to develop a more robust way of monitoring and managing the impact of obesity on the hospital system. As hospital documentation systems take the transition from paper to electronic medical records, this could present an opportunity for BMI to be directly entered and transcribed across into administrative coding in the future. While an administrative code for all patients who are obese according to BMI may not necessarily change the AR-DRG of the admission, this process has potential to ensure maximal and appropriate activity based funding is received where the case-mix has a change in DRG with co-morbid obesity assigned.

## 5. Conclusions

Three public hospitals in Queensland, Australia had prevalence rates of 30% for overweight and 32% for obesity during an audit of inpatients in 2016. Only half of the obese patients were recognised as obese by the hospital administrative system, which would have impacted funding in 3.2% of cases. While small in case numbers, extrapolated beyond a prevalence study, this has potential to impact funding on a larger scale. However, as hospital funding is currently based on administrative data, changes are required to ensure appropriate funding is provided to recognise the complexity associated with treating inpatients with obesity. Opportunities exist to strengthen ties between clinicians and coders to optimise clinical coding to better reflect patient complexity, including exploring ways for clinicians to more clearly highlight obesity and impact on patient care to support appropriate coding.

## Figures and Tables

**Figure 1 healthcare-08-00334-f001:**
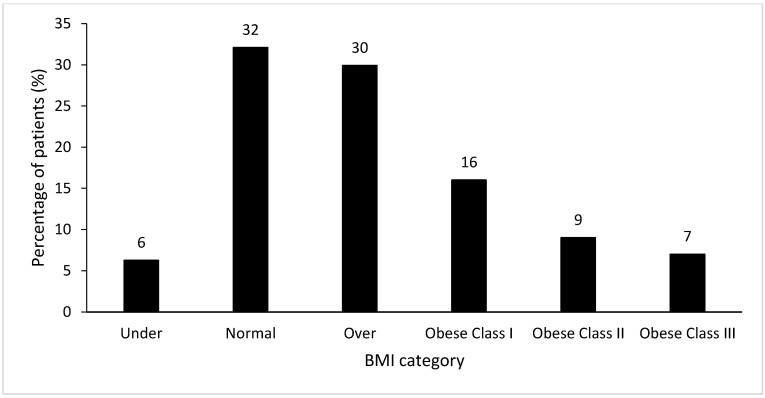
Prevalence of BMI category in the combined hospital cohort (*n* = 1327). The categories used were the WHO BMI criteria19, defined as follows: underweight (BMI < 18.50 kg/m^2^), normal range (BMI 18.50–24.99 kg/m^2^), overweight (BMI 25.00–29.99 kg/m^2^), obese class I (BMI 30.00–34.99 kg/m^2^), obese class II (BMI 35.00–39.99 kg/m^2^), and obese class III (BMI ≥ 40.00 kg/m^2^).

**Table 1 healthcare-08-00334-t001:** Key clinical characteristics of the cohort.

	Site A (*n* = 341)	Site B (*n* = 565)	Site C (*n* = 422)	All (*n* = 1327)	Comparison between Sites
Age, years Mean (SD)	58.4 (19.1)	60.4 (18.2)	63.1 (20.8)	60.7 (19.3)	*p* < 0.01
Sex, male *n* (%)	179 (52%)	344 (61%)	235 (56%)	758 (57%)	*p* = 0.037
BMI, kg/m^2^ Mean (SD)	27.6 (7.8)	28.7 (8.3)	28.1 (9.3)	28.2 (8.5)	*p* = 0.170
BMI category, *n* (%)					
Underweight (<18.5 kg/m^2^)	27 (8%)	27 (5%)	29 (7%)	83 (6%)	
Normal (18.5–24.9 kg/m^2^)	113 (33%)	167 (30%)	146 (35%)	426 (32%)	
Overweight (25–29.9 kg/m^2^)	98 (29%)	186 (33%)	113 (27%)	397 (30%)	
Obese Class I (30–34.9 kg/m^2^)	57 (17%)	86 (15%)	70 (17%)	213 (16%)	
Obese Class II (35–39.9 kg/m^2^)	25 (7%)	56 (10%)	34 (8%)	115 (9%)	
Obese Class III (≥40 kg/m^2^)	21 (7%)	43 (8%)	30 (7%)	93 (7%)	

**Table 2 healthcare-08-00334-t002:** Percentage of patients coded for obesity by BMI category.

	Underweight	Normal	Overweight	Obese Class I	Obese Class II	Obese Class III
ICD coding of obesity, *n* (%) ^1^						
Site A	0 of 26 (0%)	0 of 101 (0%)	4 of 92 (4%)	6 of 55 (11%)	10 of 23 (43%)	9 of 21 (43%)
Site B	0 of 27 (0%)	1 of 167 (1%)	10 of 186 (5%)	61 of 86 (71%)	39 of 56 (70%)	26 of 43 (60%)
Site C	0 of 28 (0%)	2 of 139 (1%)	8 of 104 (8%)	22 of 63 (35%)	15 of 32 (47%)	15 of 29 (52%)

^1^ Percentage is relative to number of patients within each BMI category at the particular site for whom coding data were available (total *n* = 1278). The categories used were the WHO BMI criteria^19^, defined as follows: underweight (BMI < 18.50 kg/m^2^), normal range (BMI 18.50–24.99 kg/m^2^), overweight (BMI 25.00–29.99 kg/m^2^), obese class I (BMI 30.00–34.99 kg/m^2^), obese class II (BMI 35.00–39.99 kg/m^2^), and obese class III (BMI ≥ 40.00 kg/m^2^).

**Table 3 healthcare-08-00334-t003:** Cost of inpatient stay and associated LOS (rounded to the nearest Australian dollar).

Total Cost ($AUD)	Underweight	Normal Weight	Overweight	Obese Class I	Obese Class II	Obese Class III
Mean	$33,344	$37,713	$42,251	$50,301	$40,356	$43,221
(SD)	($34,019)	($59,365)	($55,375)	($99,440)	($47,401)	($80,547)
Length of stay, days Mean (SD)	Underweight	Normal Weight	Overweight	Obese Class I	Obese Class II	Obese Class III
Site A	16.74 (17.96)	15.00 (14.55)	16.95 (14.73)	16.03 (14.00)	22.45 (20.49)	15.12 (21.30)
Site B	48.15 (67.00)	34.38 (52.01)	24.11 (30.63)	26.63 (31.97)	29.80 (41.60)	38.98 (51.79)
Site C	18.56 (14.37)	22.11 (26.66)	21.64 (25.41)	24.48 (31.09)	20.49 (24.40)	17.78 (27.03)

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
