# Peer review of "Underreporting of Obesity in Hospital Inpatients: A Comparison of Body Mass Index and Administrative Documentation in Australian Hospitals"

_healthcare, 2020, doi:10.3390/healthcare8030334_

Round 1

Reviewer 1 Report

Thank you for the opportunity to review this important manuscript. See below my comments. Can also be found on the manuscripts.

Line 46-Delete rate

Line 58: spending??

Line 86- delete process

Line 94- The word "this" seems not appropriate here. the reader could be kept in dark/loop on the message you are trying to pass across. pls compete the statement

Line 111-115- I think it would be good for authors to give a reasons for their exclusion and inclusion in this study. this may help those who might want to replicate the study in another zone.

Line 117-126: This shows the inconsistencies in the record if some data are missing and the alternatives are being used, since measurement are to be taking on the first patient visit. also, verbally asking the patient their weight or height may also be subjective. I don't mean to knock this already a good work done down, but this should be included as part of the limitations observed in the study.

Line 130: This looks like a repetitions

Line 164-167: underlined, should be moved to the method section. On the other hand, since the excluded data were not analyzed, it should not appear in the results.

Line 170: I guess this is a mistake? Is it supposed to be 341?? pls correct

Line 208: This statement seems to have been mentioned in the methodology section and should not be repeated.

Line 209: Start with "This study highlighted........."

Line 220: not necessary

Line 224: This statement should be moved to start a statement in Line 213. and start with something like " in order to understand the potential influence of regional factors on prevalence rates of obese patients within acute hospitals, the findings from the present study shows that...............". After this findings, the author may then compare their findings with Dennis et al study.

Limitations: it is also important that the hospital staff in charge of measurement of the anthropometric parameters do their job in order to avoid incomplete patient data. This might negatively impact the funding allocation to treat and manage obesity.

Line 248: The statement should be rephrased

Line 268: check for the right statement.

Line 276: it is important for the author to make sure the references are in line with journal requirements, both references in the list and in-text.

The manuscripts can be improved and needs to be read by the senior expert in the field for restructuring and flow.

Author Response

Reviewer 1:

Thank you for the opportunity to review this important manuscript. See below my comments. Can also be found on the manuscripts.

Line 46-Delete rate

This has been actioned. See line 46.

Line 58: spending??

Wording changed to “spent on” as per recommendations. See line 58

Line 86- delete process

Actioned per recommendations. Line 86

Line 94- The word "this" seems not appropriate here. the reader could be kept in dark/loop on the message you are trying to pass across. pls compete the statement

This sentence has been reworded to “This project endeavored to understand the significance of obesity within acute hospital admissions and how this is reflected in current coding practices, in order to develop management strategies for activity based health care delivery and resource use” (line 93-95)

Line 111-115- I think it would be good for authors to give a reasons for their exclusion and inclusion in this study. this may help those who might want to replicate the study in another zone.

Line 111 – site B excluded MH due to offsite location – this has been added. See addition to exclusion criteria set by the research team as deemed not appropriate (line 111-115)

Line 117-126: This shows the inconsistencies in the record if some data are missing and the alternatives are being used, since measurement are to be taking on the first patient visit. also, verbally asking the patient their weight or height may also be subjective. I don't mean to knock this already a good work done down, but this should be included as part of the limitations observed in the study.

Certainly, this is considered a limitation to the study and has been added in line 264-267  This now reads:

“Further limitations of this study include the variation in data collection procedures between sites and reliance on self-reported height and weight data in some circumstances, and the lack of costing data from one site due to significant transformation in records management at the time.”

Line 130: This looks like a repetitions

This has been amended to avoid repetition. 129-130

Line 164-167: underlined, should be moved to the method section. On the other hand, since the excluded data were not analyzed, it should not appear in the results.

 In methodology section we have added line 107 to discuss exclusion. Senior reviewers feel number of included/excluded belongs in the results and not the methodology section. The sentence on exclusion in the results section has been subsequently removed.

Line 170: I guess this is a mistake? Is it supposed to be 341?? pls correct

Error – amended to 341.

Line 208: This statement seems to have been mentioned in the methodology section and should not be repeated.

Senior reviewers have been unable to see this repeated in the methodology section.

Line 209: Start with "This study highlighted........."

Actioned. See line 210.

Line 224: This statement should be moved to start a statement in Line 213. and start with something like " in order to understand the potential influence of regional factors on prevalence rates of obese patients within acute hospitals, the findings from the present study shows that...............". After this findings, the author may then compare their findings with Dennis et al study.

This has been amended per recommendations.

Limitations: it is also important that the hospital staff in charge of measurement of the anthropometric parameters do their job in order to avoid incomplete patient data. This might negatively impact the funding allocation to treat and manage obesity.

This has been added as per recommendations – see line 238-241

Line 248: The statement should be rephrased

This has been rephrased.

Line 268: check for the right statement.

Actioned. Removed the error “6. Patents” – see line 269.

Line 276: it is important for the author to make sure the references are in line with journal requirements, both references in the list and in-text.

Journal requirements have been followed throughout the manuscript.

The manuscripts can be improved and needs to be read by the senior expert in the field for restructuring and flow.

Senior members of the team have been heavily involved in the review process.

Reviewer 2 Report

Well done on a most interesting paper. The main feedback is to provide greater information in your methodology on the choice of hospital sites (ease of access or ....); inclusion and exclusion criteria and treatment of underweight in your sample and your analysis.

I have a few suggestions/clarification detailed below:

Page 2 Line 81 to 83 – Repetition of abbreviation of DCL

Page 3 line 117 states that “height and weight should be documented on hospital admission, with weight documented weekly hereafter”. Please clarify is this is a hospital policy or procedure or due to guidelines of care as you state “should”.

Page 3 line 118-119 – If not documented, height was measured ….. This sentence is quite choppy – suggest to rework.

Page 3 line 131-132 – How does this statement on AR-DRG add information beyond what was written on page 3 lines 72 to 82?

Page 3 lines 146-147 states “the total actual cost of the admission for participants was obtained from the finance departments”. I’m questioning the use of the term participants and it’s appropriateness as these study patients were selected and did not consent to participate.

Page 4 lines 164-167 – At site A, 407 patients were eligible, however complete data was obtained- 66 excluded. I’m unclear if I had missed the discussion on eligibility and reasons for exclusion earlier – I double checked the method and it was not there but found it on line 167 on that states that patients were excluded if data for weight/height was incomplete. I would suggest that the discourse on eligibility and exclusion be moved to the earlier section in methodology.

Page 4 – Table 1 had data on 83 patients that were underweight (or 6.3%) but no discussion on them. While the paper is focused on obesity and management, some additional discussion is needed on those that are underweight.

Page 4 Table 2 – I would suggest that row and column total be included and provide footnote that the % are for row totals.

Page 5 Line 184 had BMI 2% (n=22) – unsure where this figure came from? Is it for site C Obese Class I according to table 2?

Page 6 Table 3 contained data on the total cost (mean and SD) and LOS for different weight categories. Again no mention in the text before or after the table on underweight category particularly given the LOS had a mean of 48 days much longer than being overweight or obese. Some discussion is needed on this

Page 6 Line 6 – needs to be reworked – the word indicate is written twice in the sentence

Page 6 – 1st paragraph in discussion on coding affecting revenue – your paper suggests that the data may exist to demonstrate or allude to this. This was not presented in this paper. Recommend that it either is presented or that is reworked. I would expect that it would be a weakness of the paper if it was not presented.

Page 6 line 233 has the coding practices at the participating hospitals are currently under review – what is the time period of currently here? Is it 2020?

Page 7 line 236-237 states that further changes to the activity based funding model would be required for the coding of obesity. This is a strong statement given the nature of the activity based funding model and how it is regulated.

Your discussion of limitations beginning on page 7 lines 238 onwards did not provide any information on the data collection process for these sites or the date of the data (2016). More is needed.

Page 7 line 268 – unsure why there is 6. Patents?

Your conclusion is very brief and needs more detail.

Author Response

Reviewer 2:

Well done on a most interesting paper. The main feedback is to provide greater information in your methodology on the choice of hospital sites (ease of access or ....); inclusion and exclusion criteria and treatment of underweight in your sample and your analysis.

I have a few suggestions/clarification detailed below:

Page 2 Line 81 to 83 – Repetition of abbreviation of DCL

Thank you – have amended this in line 83.

Page 3 line 117 states that “height and weight should be documented on hospital admission, with weight documented weekly hereafter”. Please clarify is this is a hospital policy or procedure or due to guidelines of care as you state “should”.

I have added “per hospital procedures” as this should be done across all hospitals, however, sometimes is not done in practice due to a range of possible reasons.

Page 3 line 118-119 – If not documented, height was measured ….. This sentence is quite choppy – suggest to rework.

Line 120-122 – see amended sentence structure. Now states “If not documented in the patients medical chart, height was measured with a stadiometer. Where this was not possible, height was estimated by alternative measure (i.e. demi-span, ulna-span) or verbally reported by the patient.”

Page 3 line 131-132 – How does this statement on AR-DRG add information beyond what was written on page 3 lines 72 to 82?

Agree – does not add information therefore has been deleted.

Page 3 lines 146-147 states “the total actual cost of the admission for participants was obtained from the finance departments”. I’m questioning the use of the term participants and it’s appropriateness as these study patients were selected and did not consent to participate.

Wording changed to “patients included in the study” - see line 148.

Page 4 lines 164-167 – At site A, 407 patients were eligible, however complete data was obtained- 66 excluded. I’m unclear if I had missed the discussion on eligibility and reasons for exclusion earlier – I double checked the method and it was not there but found it on line 167 on that states that patients were excluded if data for weight/height was incomplete. I would suggest that the discourse on eligibility and exclusion be moved to the earlier section in methodology.

Methodology section has been extended to include information on excluded data. Refer to methodology section

Page 4 – Table 1 had data on 83 patients that were underweight (or 6.3%) but no discussion on them. While the paper is focused on obesity and management, some additional discussion is needed on those that are underweight.

I’ve added a sentence line 211-214 acknowledging this. We did not investigate this further as not part of the study.

Page 4 Table 2 – I would suggest that row and column total be included and provide footnote that the % are for row totals.

This table has been amended by a senior reviewer. See table 2 (line 192)

Page 5 Line 184 had BMI 2% (n=22) – unsure where this figure came from? Is it for site C Obese Class I according to table 2?

This was the proportion (2% - 2 out of 1276) of patients who had been coded for obesity from hospital information management services (I’ve clarified this sentence se 184-185)

Page 6 Table 3 contained data on the total cost (mean and SD) and LOS for different weight categories. Again no mention in the text before or after the table on underweight category particularly given the LOS had a mean of 48 days much longer than being overweight or obese. Some discussion is needed on this

LOS for underweight patients was only higher than the obese at site B. I’ve added a sentence line 211-213 but do we need to add in further??

Page 6 Line 6 – needs to be reworked – the word indicate is written twice in the sentence

Page 6 – 1st paragraph in discussion on coding affecting revenue – your paper suggests that the data may exist to demonstrate or allude to this. This was not presented in this paper. Recommend that it either is presented or that is reworked. I would expect that it would be a weakness of the paper if it was not presented.

Please refer to the final paragraph of results line 210-212 “Results demonstrated that if all 94 obese patients had an additional code of “obese”, only 3 (3%) would have had a change in DRG and subsequent activity based funding”

Page 6 line 233 has the coding practices at the participating hospitals are currently under review – what is the time period of currently here? Is it 2020?

I have added “in 2020” into the text line 239.

Page 7 line 236-237 states that further changes to the activity based funding model would be required for the coding of obesity. This is a strong statement given the nature of the activity based funding model and how it is regulated.

Amended. See 251-254 for wording changes.

Your discussion of limitations beginning on page 7 lines 238 onwards did not provide any information on the data collection process for these sites or the date of the data (2016). More is needed.

This has been added per the reviewer comments. Page 7 line 268 – unsure why there is 6. Patents?

Error – deleted. Line 269

Your conclusion is very brief and needs more detail.

Further comment has been added to the conclusion from senior members – see line 286-294

Round 2

Reviewer 1 Report

I would like to thank the authors for comprehensive response to my comments. I am satisfied with all the responses provided.